# The Impact of Germinated Chickpea Flour Addition on Dough Rheology and Bread Quality

**DOI:** 10.3390/plants11091225

**Published:** 2022-04-30

**Authors:** Denisa Atudorei, Olivia Atudorei, Georgiana Gabriela Codină

**Affiliations:** Faculty of Food Engineering, Stefan cel Mare University of Suceava, 720229 Suceava, Romania; denisa.atudorei@outlook.com (D.A.); olivia.atudorei@outlook.com (O.A.)

**Keywords:** germinated chickpea flour, white wheat flour, dough rheology, bread quality

## Abstract

The research focused on the effect of germinated chickpea flour (GCF) in a lyophilized form on dough rheology, microstructure and bread quality. The GCF addition levels in refined wheat flour with a low α-amylase activity were 5%, 10%, 15% and 20%, up to an optimum falling number value of the mixed flour. Generally, the dough rheological properties of water absorption, tolerance to mixing, dough consistency, dough extensibility, index of swelling, baking strength and loss tangent (tan δ) for the temperature sweep test decreased with the increased level of GCF addition, whereas the total volume of gas production and G′ and G″ modules for the temperature sweep test increased. Dough microstructure analyzed by epifluorescence light microscopy (EFLM) clearly showed a change in the starch and gluten distribution from the dough system by an increase in protein and a decrease in starch granules phase with the increased level of GCF addition in wheat flour. The bread physical characteristics (loaf volume, porosity, elasticity) and sensory ones were improved with up to 15% GCF addition in wheat flour. The bread firmness increased, whereas the bread gumminess, cohesiveness and resilience decreased with increased GCF addition in wheat flour. The bread crust and crumb color of the bread samples become darker with an increased GCF addition in the bread recipe.

## 1. Introduction

Improving food in terms of quality and nutrition has always been in high demand. Kotsiou et al. [1] have reported that an addition from 10 to 20% chickpea flour had led to an improvement in the bread-making process. According to them, the addition of chickpea flour increased the value of water absorption. They also reported that an addition of 10% chickpea flour increased the dough stability compared with the control sample. The substitution of wheat flour by 10% chickpea flour did not negatively affect the quality of the bread. The tasters enjoyed the bread samples with the addition of chickpea flour, reflecting they were attractive in terms of color, taste and texture. Compared with other legumes, chickpea protein has higher values for the following parameters: foaming capacity, stability and thermal expansion. Chickpea protein also has a higher emulsion stability index, which makes chickpea flour suitable for use as an ingredient in bread making [2]. Some authors have pointed out that the addition of chickpea flour in the bread-making recipe increased the extensibility value of dough [3]. Therefore, in view of the considerations already mentioned, the potential of chickpeas as an ingredient in the bread-making process is worth highlighting in terms of both the dough rheology and the bread quality. This potential is worth evaluating as an addition of chickpea flour in its native form but also in other forms, such as germinated, fermented and roasted. The addition of chickpea flour in various forms to the bread-making recipe is of interest in terms of both the rheology of the dough and the quality of the bread, as well as in terms of its nutritional value, to remedy various shortcomings.

Nowadays, consumer demand for foods with high nutritional value is constantly growing [4]. Food producers are trying to enrich regular foods which are low in some nutrients with different ingredients that may improve their nutritional content. Bread is one of the most consumed food products worldwide, especially bread obtained from refined flour [5], which is nutritionally poorer than those obtained from wholemeal flour [6]. Bakery products reflect the nutritional values of the ingredients used, making them important sources of carbohydrates, proteins, mineral salts and B vitamins [7]. Legumes are nutritionally complementary to white wheat flour, coming as a supplement to the content of fibers, proteins, minerals and vitamins [8]. Legumes contain proteins with balanced nutritional profiles and are rich in phytochemicals that have antioxidant properties and contain a significant amount of dietary fiber [9]. The protein content of legumes varies between 20 and 40%. Legumes contain an amount of fiber in the range of 14-33%, depending on the legume type. In the category of vitamins and minerals, legumes contain high amounts of thiamine, riboflavin, folic acid, calcium, iron and zinc [10].

The World Health Organization emphasizes that vegetables are important foods for health because they are basic sources of protein and other essential nutrients [11]. Although nutritionists recommend eating legumes as often as possible, their consumption worldwide remains below the recommended limit, even today. Therefore, it is desirable to incorporate them into foods that are more commonly consumed [12]. Among legumes, chickpeas (*Cicer arietinum* L.) occupy an important place due to their significant nutritional profile. Regarding the nutritional aspect, chickpeas are distinguished by the fact that they contain valuable nutritional compounds: carbohydrates, proteins, minerals, unsaturated fatty acids, isoflavones, dietary fiber, 18 different types of amino acids (8 essential amino acids) and lipids [13]. Medical studies have shown that the consumption of chickpeas has important contributions in the treatment of diseases such as cancer, osteoporosis, cardiovascular disease and hyperlipidemia [14]. Numerous other studies have also focused on chickpea composition and nutritional utilization, highlighting its positive aspects related to human nutrition and the possibility of being used as an ingredient to obtain functional foods [15,16]. Thus, using chickpea flour in bread recipes will produce bakery products with high nutritional value. In addition to the nutritional aspects, it must be mentioned that the cultivation of this vegetable is not complex or resource-intensive, in the sense that it can be grown in areas with low rainfall because it is a drought-resistant plant. Chickpea cultures are not expensive, being the second most cultivated legume worldwide, after beans. However, chickpea contains some antinutritional factors, such as tannins, phytic acid and unpleasant flavor compounds [17] that may be minimized by subjecting it to germination. Germination is a process that generally refers to the process of sprouting grains in order to transform them into plants [18]. As studies in the field have reported, the germination process brings with it changes in the biochemical, sensory and nutritional profile of grains subjected to this process [19], improving the quality of legumes. According to our previous study [20], chickpea germination leads to a decrease in lipid and carbohydrate content and to an increase in protein content of up to 21.1%. The amount of sodium, magnesium, iron and zinc also increased, indicating improvements in nutritional value. Additionally, some authors have pointed out that the germination process had a special effect on the phenolic compounds and bioactive ingredients of the seeds subjected to it. For example, Munarko et al. [21] have shown that germination increased the amount of γ-aminobutyric acid, total phenolic and flavonoid content and the antioxidant activity. At the same time, Wu et al. [22] highlighted in a study that germination led to the biosynthesis and accumulation of lignans and phenolic acids and led to the improvement of antioxidant activities. However, the outcome was different depending on the variety of seeds subjected to the germination process. Germination is of interest in the food industry because the enzymes in the grains are activated during the germination process, and it may be used in different technological processes to obtain various improved food products from the quality point of view [23]. In bread making, the increased amount of amylases and proteases from the germination process of chickpea may be beneficial in improving the technological process and bread quality obtained from a strong wheat flour with a low α-amylase activity.

To our knowledge, there are currently few studies on the possibility of using chickpea flour in germinated form. In general, the studies have been more focused on using the addition of non-germinated chickpea flour in bread making. A study by Mohammed et al. [24] showed that the addition of germinated chickpea flour in the bread-making recipe had the effect of increasing the water absorption and dough development time and reducing the extensibility of dough and the resistance to deformation. Regarding the quality of the bread, the authors concluded that an addition of 10% did not negatively affect the quality parameters: volume, color of the bread crust and crumb, the porosity of the crumb and the textural parameters of the bread. Additionally, Kotsiou et al. concluded that the addition of roasted chickpea flour up to a level of 10% in wheat flour did not have a negative effect on the dough’s rheological properties and bread quality [1]. Atudorei et al. [25] highlighted that a combination of germinated chickpea flour (GCF) and germinated lupine has a positive effect on dough rheological properties up to a certain level of 8.57% in the case of GCF addition to wheat flour. The aim of this study was to highlight the effect of the addition of GCF on the rheological properties of the dough and on the quality characteristics of the bread obtained from refined wheat flour of a strong quality and a low α-amylase activity. Additionally, the impact of GCF addition on dough microstructure had also been studied. The addition of GCF is desirable because it is expected to increase the quality of the finished product, since the enzymes activated in the germination stage may improve the technological characteristics of the mixed flour.

## 2. Results

### 2.1. Flour Characteristics

For white wheat flour, the following data were obtained: 14.6% moisture, 0.66% ash content, 12.3% protein, 1.12% fat, 30.4% wet gluten, 3 mm gluten deformation index, 350 s falling number value. For the germinated chickpea flour, the physicochemical characteristics values were 3.6% ash, 21.1% protein, 5.2% fat. According to the data obtained, the wheat had a low α-amylase activity and had strong qualities for bread making [26]. The GCF had high protein content.

### 2.2. Dough Rheological Properties

#### 2.2.1. Dough Rheological Properties during Mixing and Extension

The dough rheological data during mixing and extension are shown in Table 1. According to the Consistograph values the GCF addition led to a significant decrease (*p* < 0.05) in water absorption (WA) and tolerance to mixing (Tol) when high levels of GCF were incorporated in the dough recipe. Dough consistency after 250 s (D250) and 450 s significantly decreased (*p* < 0.05) up to 15% GCF addition in wheat flour, after which their values increased.

Table 2 shows the Alveograph data for the dough samples with different levels of GCF addition in wheat flour. According to the data obtained, it may be seen that dough extensibility, index of swelling and baking strength decreased by GCF addition in wheat flour up to the 15% level, whereas up to 20% addition, these values insignificantly increased (*p* < 0.05). Generally, the maximum pressure slightly increased when GCF was incorporated in the dough recipe.

#### 2.2.2. Dough Rheological Properties during Fermentation and Falling Number Values

Table 3 shows the Rheofermentometer data for dough samples with different levels of GCF addition in wheat flour. As can be seen, the value of the parameter H’m (maximum height of gaseous production) increased at the addition levels of 5% and 10% GCF in the wheat flour. Starting with an addition level of 15% GCF to wheat flour, its value decreased, being lower for the sample with 20% GCF than for the control sample. The same trend may be noticed for VT (total CO_2_ volume production). Regarding the VR parameter (volume of the gas retained in the dough at the end of the test), it may be noticed that, compared with the control sample, a lower value was recorded for the dough samples with addition levels of 10% and 20% GCF to wheat flour. The CR (retention coefficient) parameter was higher than in the case of the control sample at an addition level of 20% GCF to wheat flour, otherwise its value decreased due with the GCF addition. The FN (falling number) parameter values significantly decreased (*p* < 0.05) as the GCF addition level to wheat flour increased.

#### 2.2.3. Dough Fundamental Rheological Properties

Figure 1 shows the diagrams obtained in the case of frequency sweep tests, from which it can be seen that the three parameters, the storage modulus, the loss modulus and the loss tangent, are largely determined by frequency. As can be seen from the figure, the trend for the storage modulus (G′) is positioned on the axis above that for the loss modulus (G″). The graph also shows that the values for the two parameters increased with the increased level of the GCF addition to the dough matrix. In the case of the tan δ (loss tangent) parameter, it may be seen that it increased in a frequency-dependent manner. Additionally, the ratio of viscous and elastic components was less than 1, regardless of the level of GCF addition to wheat flour.

Figure 2 shows the G′, G″ and tan δ evaluation with the temperature increased. According to their evaluation, it may be seen that, up to a certain temperature, their value decreased probably due to the proteins denaturation, then increased because the starch gelatinized. Generally, all the dough samples with GCF addition to the wheat flour presented with lower values for G′ and G″ and higher ones for tan δ compared with the control sample.

### 2.3. Dough Microstructure

Figure 3 shows the microstructure of the dough samples, with images captured using EFLM. The red color highlights the proteins, highlighted by rhodamine B, and the green color highlights the starch granules, highlighted by fluorescein. Thus, from Figure 3, it may be observed that the addition of GCF resulted in obtaining images more intensely colored in red and less colored in green. This means that, with the increase in the percentage of GCF in wheat flour, the amount of protein increased and the amount of starch decreased.

### 2.4. Bread Quality Evaluation

#### 2.4.1. Bread Physical Characteristics

The physical characteristics of the bakery products are ones of the most important parameters because they are directly related to the perception and choice of consumers. Evaluating the influence of the addition of GCF on the physical properties of the bread (loaf volume, porosity, elasticity) is essential because it has a direct influence on consumer preference. The physical bread characteristics are shown in Table 4. According to the data obtained, it can be noticed that the addition of GCF changed all the physical parameters of the bread samples. A maximum of 15% addition was effective due to the fact that the values of the three physical characteristics of the bread samples, specific volume, porosity and elasticity, had significantly (*p* <0.05) increased. However, an addition of more than 15% would not be desirable because the bread physical characteristics of the bread samples were lower compared with the control sample.

#### 2.4.2. Color Parameters of Bread Samples

Consumer acceptability is also related to the color of the bread samples, both the color of the crust and of the bread crumb. Table 5 shows how the addition of GCF influenced the color parameters of the bread samples. According to our data, it can be seen that both the crust and the crumb color of the bread samples were significantly (*p* < 0.05) influenced by GCF addition in the wheat flour. Thus, the lightness parameter (L*) decreased in value, which means that the samples darkened in color due to the GCF addition. In the case of the a* parameter, it can be observed that it increased in the case of both the crust and the bread crumb. Thus, it can be concluded that the samples with GCF addition had a more pronounced red tint. Regarding the b* parameter, its value had increased, which means that the GCF addition had contributed to the intensification of the yellow hue of the bread samples.

#### 2.4.3. Texture Profile Analysis of Bread Samples

Table 6 shows that the addition of GCF increased the values of the firmness parameter for each level of addition. The value of the gumminess characteristic decreased at 5–15% GCF addition in wheat flour. At an addition level of 20% GCF, the value of this parameter increased. For cohesiveness and resilience, lower values were recorded for samples with GCF incorporated in the bread recipe compared with the control sample. Evaluation of the influence of the GCF addition in wheat flour on the bread textural parameters is important because it indicates how the bread sample is perceived by the consumers when it is chewed, which influences consumer perceptions of the quality of the product.

#### 2.4.4. Crumb Structure of Bread Samples

The crumb structure of the bread samples with different levels of GCF in the bread recipe are shown in Figure 4. As can be seen from the images obtained, the GCF addition led to a change in pore size and density. At low levels of GCF addition, the bread crumb presented larger pores, and at high GCF levels, GCF addition in the bread recipe let to a decrease in the pores’ density.

#### 2.4.5. Sensory Analysis of Bread Samples

Figure 5 shows how the addition of GCF influenced consumer perceptions of the assessed organoleptic properties (appearance, color, taste, smell, texture, aroma and overall acceptability). From this figure, it can be seen that the GCF addition influenced each organoleptic property separately. Compared with the control sample, the sample with the addition of 15% GCF was the best appreciated by the participants in the tasting session, and the sample with the addition of 20% was the least appreciated. In view of these considerations, it can be stated that an addition of 15% is the optimal percentage for consumer satisfaction from a sensory point of view.

## 3. Discussions

### 3.1. Dough Rheological Properties

#### 3.1.1. Dough Rheological Properties during Mixing and Extension

The decrease in the value of the WA parameter with the increased level of GCF addition in wheat flour may be explained by the fact that the germination process produces a series of changes in the composition profile of the grains. During germination, the activated enzymes break proteins down into amino acids and peptides, and parts of the proteins and starch are used to develop the component parts of the sprouts, decreasing the amount in their content and causing a decrease in the ability of chickpea flour to emulsify and to foam [27]. More, the dextrins formed present a lower water binding capacity [28]. Similar results, in the sense of decreasing the value of the water absorption parameter, had also been reported by Xing et al. [29] for dough samples with chickpea-protein-enriched fraction addition. According to them, the WA decrease was due to the fact that white wheat flour gluten absorbs more water than other proteins. According to Kotsiou et al. [1], the decreased WA for dough samples with chickpea flour addition was due to the fact that chickpeas contain a higher amount of protein and fiber than white wheat flour. Therefore, the fibers and proteins can contribute extensively to the WA of the mixed flour [25]. The enzyme content of the mix of ingredients had an important influence on the water absorption index (WA) and, implicitly, on the rheology of the dough. This aspect has been evaluated in various studies. For example, Yang et al. [30] highlighted that a mixture of enzymes, such as xylanase, glucose oxidase and cellulase, improved the dough rheological properties more than the case when only one type of enzyme was added in the wheat flour. The enzymes mixture positively affects the extensibility, tenacity, and stability of the dough. This was due to the fact that there was a reduction in the free sulfhydryl in the dough matrix, which led to an increase in water-extractable arabinoxylan content. Considering the effect of the addition of germinated chickpea flour, it can be stated that it contains a mix of enzymes that were activated during the germination stage, which led us to conclude that each enzyme had a complementary effect on improving the dough rheology: a more intense effect than if only one type of enzyme had been added to the dough recipe. For example, Guardado-Félix et al. [31] explained that germinated chickpea flour is rich in α-amylase and protease activity, which has an effect on the gluten network. Therefore, it has an effect on the rheology of the dough, as well as on the quality of the bread. The increased tolerance to mixing up to an addition level of 5% and 10% GCF is explained by the fact that the GCF addition produced an increase in the stability of the gluten network. These data are in agreement with other authors results, which concluded that, when low levels of GCF were added to wheat flour, the dough stability increased [32,33]. However, the increased enzymatic activities of the mixed flours had an important role in the dough’s rheological properties during mixing [25]. The decreased value of the tolerance to mixing at high levels of GCF in wheat flour may be explained by the fact that the activated amylases in the germination stage produced a hydrolysis of starch [34], which coincided with the increased amount of maltose in the dough system. Additionally, the decreased amount of gluten in the dough matrix may result in a decrease in the stability of the dough. A similar trend of decreasing the dough stability was also obtained by Millar et al. [27] for dough samples with germinated pea flour addition in wheat flour. The decrease in dough consistency (after 250 s and after 450 s), in the addition range of 5%–15% GCF to wheat flour, may be due to the specific compounds from the composition of chickpeas (proteins, fibers, carbohydrates) that interfere with gluten in the kneading stage, resulting in an increase in viscosity and a decrease in dough consistency. The increase in dough consistency values in the case of a 20% GCF addition is explained by the fact that the continuous degradation of starch granules in the case of a higher amount of enzymes from the dough system led to a decreased dough viscosity, which led to an increase in dough consistency [35,36].

A slight increase in value of the maximum pressure (P) parameter indicates that the GCF addition resulted in a very slightly firmer dough, which was significantly higher (*p* < 0.05) compared with the control sample, but only to a level of 15% GCF addition to the dough recipe. The explanation behind this behavior is that sprouted chickpea flour contains a higher amount of protein, which led to firmer dough. Additionally, the fibers in the chickpea flour interfere with the gluten proteins from white wheat flour, resulting in dough with a slightly higher strain resistance. The decreased value of the dough extensibility index (L) as the GCF addition level increased can be attributed to several considerations. First of all, by the increasing amount of GCF in the dough recipe, the amount of gluten from the dough matrix decreased, which leads to a dough with a lower expansion capacity. Additionally, with the increase in the GCF addition level, the amount of sulfhydryl and thiol groups found in the mix increased, which had the effect of intensifying the oxidation reactions in the dough during the kneading process. At the same time, with the addition of chickpea flour in the dough matrix, a competition for water may take place between the fibers and proteins from the chickpea and wheat flours. Similar results were reported in a study by Mohammed et al. [24] for dough samples with different levels of chickpea flour addition in wheat flour. The index of swelling parameter (G) also decreased, up to a maximum level of 15% GCF, after which its value insignificantly increased (*p* < 0.05). This decrease may be attributed to a decrease in the amount of gluten from the dough system. The decrease in the Alveograph value baking strength (W) with the increased level of GCF in the dough recipe may be explained by gluten dilution, when the chickpea flour, which has no gluten in its composition, was added to the wheat flour. Furthermore, the proteins from GCF compete with the gluten proteins of the white wheat flour, which causes a delay in the formation of gluten. The configuration ration of the Alveograph curve highlights the previous trend of the Alveograph values of maximum pressure and dough extensibility. The increase in the value of the P/L ratio was mainly due to the strong interactions that take place in the dough matrix between the fibers from chickpea flour and the proteins from wheat flour [37].

#### 3.1.2. Dough Rheological Properties during Fermentation and Falling Number Values

The increase in the value of the parameter H’m (maximum height of gaseous production), compared with the control sample, up to an addition level of 5% and 10% GCF, respectively, may be explained by the activity of α-amylase enzyme, which was activated during the germination stage and which had the effect of increasing the amount of fermentable sugars [38]. Maltose was used by the yeast during the fermentation stage, which had the effect of releasing a larger amount of carbon dioxide. The results of this study are consistent with another study conducted by Marti et al. [35], who highlighted the fact that the addition of sprouted wheat flour increased the value of this parameter. The positive effect of α-amylase on the leavening stage of the dough had been also highlighted in other study [39]. However, up to addition levels of 15% and 20%, respectively, of GCF to wheat flour, the value of the H’m parameter decreased, this being explained by a lower capacity of the dough to retain the gas in the system, resulting in the proofing stage. This behavior was due to the fact that the addition of GCF into the system has a gluten-diluting effect, weakening the gluten network. Thus, the dough’s capacity to lose gases became higher. The fact that the value of the VT parameter (total CO_2_ volume production) was higher than in the case of the control sample, at the addition level of 5%, 10% and 15% GCF to the dough, means that the production of carbon dioxide in the fermentation stage had been intensified by the increased amounts of mono- and disaccharides due to the increased enzymatic activity as substrates for yeast fermentation [40]. However, it was noted that, at higher addition level of 20% GCF, the capacity of the dough structure to retain gases in the dough system was reduced. This can be explained by the fact that the activity of the enzyme α-amylase led to a decrease in the capacity of the dough to retain carbon dioxide and led to an increase in the permeability of the dough to lose gases, because there was a weakening of the gluten structure due to the starch hydrolysis [35,41]. Moreover, the protease enzyme, that had also been activated during germination, had an effect on peptide bonds, hydrolyzing them, which coincided with the destruction of the protein network and the decrease in the dough’s ability to retain carbon dioxide in the system [35]. The fact that the samples with the addition of GCF showed a decreasing value of the falling number parameter, as the addition level of GCF increased, indicates that the activity of the enzyme α-amylase was more and more intense. The results of this study are in agreement with the data reported by Lazo-Vélez et al. [42], which showed that, in the case of an addition of GCF in the amounts of 150 g/kg and 300 g/kg, the value of the falling number decreased compared with the control sample.

#### 3.1.3. Fundamental Rheological Properties of Dough

According to the data obtained, the storage modulus was higher than the loss modulus, which indicates that the viscous properties of the dough were less prominent than the elastic ones. This conclusion is due to the fact that G′ is an indicator of the elasticity of the dough, and G″ describes its viscous properties. The increase in the value of the two parameters due to the addition of 15% and 20% GCF to wheat flour, compared with the control sample, indicates that the GCF improved the viscoelasticity of the dough. Similar results were obtained by Kotsiou et al., who reported that the addition of roasted chickpea flour led to a predominance of an elastic-like behavior [1]. The influence of the addition of GCF on the rheological parameters can be explained by the fact that it contains a higher amount of protein and dietary fiber compounds that interfere with the gluten network, which caused a reduction in gluten plasticity and led to a strength effect in the dough [1]. The fact that the elasticity of the dough improved with the GCF addition is desirable because good elasticity of the dough leads to a good ability to form a gas network and to retain a bubble structure [27,43]. According to Yang et al. [44], germination had the effect of reducing the values for the storage modulus and the loss modulus in the case of buckwheat and pea. The same trend was also reported by Zhang et al. [45], who highlighted the influence of maltohexaose-producing α-amylase on dough rheology. The tan δ value increased due to the GCF addition, with means that there was an increase in the ratio of the elastic structure in the dough matrix. The tangent loss parameter had a value less than 1, which indicated that the dough had solid-like behavior for all the samples analyzed.

The temperature sweep graphs illustrate the effect that the GCF addition had on the rheological characteristics of the dough, taking into account the influence of temperature. Figure 2 show that the samples with GCF addition had a higher paste temperature than the sample without GCF addition. The results of this study are in agreement with other studies, which reported that the use of germinated brown rice, oats, and quinoa flours [46,47] had a similar effect. A higher pasting temperature with the increased level of GCF addition can be explained by the fact that enzymes that were activated in the germination stage had preferentially hydrolyzed the amorphous region, which coincided with the formation of a more rigid crystalline structure [48]. Due to the gelatinization process, as the temperature increased, the viscoelasticity of the dough also increased. However, the starch degradation process was more intense in the case of samples with GCF addition due to the α-amylase, which was activated during the germination stage. Thus, the samples with GCF addition showed a higher gelatinization capacity, which caused the values of the viscoelastic modulus to decrease. Furthermore, studies have shown that germination increased the amount of soluble dietary fibers due to the fact that the partial conversion of some insoluble dietary fiber into soluble dietary fiber may take place during the germination process, affecting the rheology and structure of the dough. The soluble fiber amount increased the competition for water between the soluble fiber and the gluten protein, with effects in the dough matrix [48]. At the same time, the value for tan δ was higher in the samples with the addition of GCF when the dough was in the starch gelatinization phase. In the optimal activity range of α-amylase, which acts on starch, the viscoelasticity of the dough decreased more and more as the GCF addition level increased because of a higher amount of α-amylase in the dough matrix.

### 3.2. Dough Microstructure

The images obtained by EFLM shown that GCF addition, especially at high levels, significantly changed the spatial distribution of starch and proteins in the dough network. Thus, in the case of the control sample, the existence of a well-defined starch structure was observed. Protein granules were scattered between the starch granules. Images with the addition of GCF showed a higher amount of protein (red areas) and less starch (green areas) with the increased level of GCF incorporated in the dough recipe. This was to be expected because GCF contains a higher amount of protein than white wheat flour. At higher amounts of GCF, the starch granules are those that appear to be diffused into the protein mass. Similar results were obtained in a previous study by Dabija A. et al. [49], which highlighted that the addition of yellow pea flour in wheat white flour led to an intensification of red areas and a decrease in green ones.

### 3.3. Bread Quality Evaluation

#### 3.3.1. Bread Physical Characteristics

An improvement in the physical characteristics of the bread samples was observed up to the 15% level of GCF addition to the wheat flour. This may be due to the chickpea enzymes that were activated during the germination process. The activation of these enzymes brought with it a number of other changes in the chickpea profile. More, once those enzymes were incorporated into the dough network, they worked further and had several aspects to their activity. Chickpea enzymes (amylases) acted on the starch which resulted in an increase in the number of fermentable sugars [50]. A higher amount of mono- and disaccharides increased the amount of carbon dioxide produced. The baker’s yeast, *Saccharomyces cerevisiae*, used these fermentable sugars in its activity and the yeast activity resulted in carbon dioxide [40]. Thus, due to the higher content of fermentable sugars, the amount of carbon dioxide released into the system automatically increased, which had the effect of increasing the specific volume of the bread samples. Similar results have been reported by Marti et al. [35], which concluded that the addition of sprouts in the bread-making recipe did not had a negative influence on the final products. The decreased volume of bread samples at 20% GCF addition can be explained by a higher amount of enzymes that led to a more intense hydrolysis of the starch network, with a negative effect on the dough’s ability to retain gases formed during fermentation [41]. At the same time, proteases that have been activated during sprouting lead to a hydrolysis of peptide bonds, which may lead to a disruption of the protein network and to a decrease in the dough’s ability to enclose air [35]. At the same time, the decrease in the bread loaf volume due to the GCF addition can also be attributed to the decrease in the amount of gluten, which led to a decrease in the viscoelastic properties of the dough and its ability to retain gases in the dough system. The addition also resulted in a disruption of the gluten network because chickpea contains a higher amount of fiber than white wheat flour [51]. Similar data were reported by other studies, which concluded that a high amount of fibers had the effect of reducing the loaf volume of bread. This effect was due to the fact that fibers caused a weakening effect on the dough network by creating a disruption in the gluten network, which could no longer retain the gas formed during the fermentation process [52].

Regarding the porosity of the samples with GCF in the bread recipe, it was observed that, up to a maximum of 15% GCF addition, this parameter was improved compared with the control sample. The positive influence of the GCF addition on porosity may be a consequence of the enzymes that were activated during chickpea germination. The amylases act on the starch from the dough system forming fermentable sugars, which lead to an improvement of baker’s yeast activity. This may lead to higher carbon dioxide and higher porosity for bread with GCF addition than for the control sample [40,53]. Similar results were reported by Marti et al. [35] for bread samples with germinated wheat flour addition, which presented a better porosity even when a high level of 50% germinated flour was added to the bread recipe. However, a decrease in porosity once the 15% GCF addition level was exceeded can be explained by the fact that the dough did not have the same capacity to retain the released carbon dioxide in the system. The addition of up to 15% GCF to the bread recipe also increased the elasticity values of the bread samples. This may be explained by the fact that the α-amylases that were activated in the chickpea germination stage had the effect of improving the elasticity of the bread crumb [45]. The decrease in crumb elasticity at 20% GCF addition can be explained by the fact that the amount of gluten in the dough network decreased to an excessive extent.

#### 3.3.2. Color Analysis of Bread Samples

The specific color of bakery products is given by the Maillard reaction especially. The darkening of the bread samples with the addition of GCF may be explained by the fact that the GCF contains simple amino acids and sugars. These sugars and amino acids were formed due to enzymes activity, which increased the amount of enzymes that acted on starch and proteins in the germination stage. These sugars and amino acids led to the intensification of the Maillard reaction, with an effect on the color of the bread samples [50,54,55]. The decrease in the lightness in color of the samples due to the addition of GCF may also be explained by the higher amount of flavonoids and phenolic compounds of the GCF flour [31]. These compounds are related to the oxidation processes which occurred in the baking stage of bread making [56]. The increase in the reddening tendency of the bread samples with GCF addition may also be attributed to the more intense Maillard and browning reactions that occurred in the bread-making process. The yellow hue may be due to the specific pigments from the chickpea composition [36]. Similar results were obtained by Guardado-Félix et al. [36], who concluded that replacing wheat flour with 15% GCF lead to a darker color of the samples and to a specific yellowish hue. Additionally, the same trend of sample color darkening due to the addition of legume flours has been reported in other studies [57,58].

#### 3.3.3. Texture Profile Analysis of Bread Samples

According to our data, the addition of GCF increased the values for the firmness parameter with increased levels of GCF in the bread recipe. Similar results have been reported by others when roasted chickpea flour was added to wheat flour [32]. Additionally, other studies concluded that the addition of legume flours, at 10-20%, had the effect of increasing the firmness of the bread crumb [59,60,61]. This may be explained by the fact that legumes contain a higher amount of amylose than wheat grains. However, although the value of the firmness parameter had increased due to the GCF addition, the alpha-amylase that was activated during the chickpea germination process may increase the shelf life of the final product [62]. The increase in the firmness of the bread can also be explained by the fact that legume flours contain a higher amount of fiber than white wheat flour. The insoluble fibers from their content may cause a disturbance in the process of forming the gluten network, which may lead to bread with higher firmness [63]. Ouazib et al. [64] highlighted that the addition of 20% GCF significantly increased the hardness of bread. They explained that this increase is due to the activity of the proteases that were activated during the germination process, which hydrolyzed the gluten proteins, leading to an effect of weakening and disrupting the gluten network [36]. Because bread of good quality has a lower value of gumminess [65], it may be concluded that bread with GCF addition has an improved quality from this point of view. Gumminess may be influenced by the ability of the ingredients to absorb water. The decrease in the cohesiveness value by GCF addition may be explained by the reduction in water absorption from the dough matrix, which led to a decrease in the moisture content of the bread crumb [27]. The decrease in the resilience parameter with GCF addition may be due to the enzymes activity which increased during the chickpea germination stage [66]. A similar trend of the resilience value was also reported by Millar et al. [27], when pea germ flour was added to the bread recipe.

#### 3.3.4. Crumb Structure of Bread Samples

The porosity of the bread was mainly influenced by the carbon dioxide that was released during the proofing and baking stage of bread making, but was also influenced by the ability of the dough to retain the released gas formed. An uneven pore distribution may be explained by the fact that the GCF addition had led to a weakening effect on the gluten network and to a decrease in the dough’s ability to retain the carbon dioxide formed. The larger pore size can be explained in terms of the enzymatic activity that led to the coalescence of the gas cells [67]. A similar trend, in the sense that the size of the pores had increased, was also reported by Cardone et al. [68] for bread samples with sprouted wheat addition to wheat flour. However, the high amounts of enzymes from the dough samples with GCF addition may present a positive effect on porosity due to their breaking effect on starch with the formation of fermentable sugars, which helped the yeast to release more gases and to improve the bread porosity [11].

#### 3.3.5. Sensory Analysis of the Bread Samples

Generally, compared with the control sample, the addition of GCF had the effect of improving the organoleptic characteristics of the bread samples. This improvement may be explained by the beneficial role of the germination process, in the sense that the activation of chickpea enzymes had the effect of improving the porosity, elasticity and loaf volume of the bread samples, which had also a positive impact on bread consumers. Regarding the taste and aroma of the bread samples, it can be concluded that they were positively influenced by the GCF addition, except for the sample with the highest GCF addition level, probably due to the germination process, which had the effect of improving the sensory profile of chickpeas by increasing the sweetness [69]. The improvement of the flavor profile of the samples with GCF addition may also be due to the intensification of the Maillard reaction. The enzymes from GCF may lead to an increase in the amount of simple sugars and amino acids, through their activity [11]. The sample with 20% GCF addition was less appreciated, probably due to the fact that higher levels of GCF may lead to bread samples with a taste and aroma specific to chickpeas, but also due to the fact that, in the dough matrix, the amount of gluten decreased significantly, which had a negative effect on bread quality. Similar results were reported by Al-Ansi et al. [70], who concluded that using some germinated ingredients in a bread recipe may improve the sensorial quality of the bread produced.

## 4. Materials and Methods

### 4.1. Materials

In order to prepare the bread samples, white wheat flour was used. The flour was purchased from the company S.C. Dizing S.R.L., from Brusturi, Neamț, România. Germinated chickpea flour (GCF) was obtained from chickpeas (Cicer aretinium L.) cultivated in România, which had not been genetically modified. The chickpeas were germinated at a temperature of 25 °C, in dark conditions and at a relative humidity of 80%. Germination was stopped after four days. The germination process was described in our previous studies [58,71]. Chickpea sprouts were then subjected to the lyophilization process, using a Biobase, BK-FD12 (Jinan, China) lyophilizer. The parameters for the lyophilization process were as follows: pressure of 10 Pa, for 24 h, temperature of −50 °C. A laboratory mill 3100 (Perten Instruments, Hägersten, Sweden) was used to grind the chickpea sprouts. The white wheat flour and GCF were analyzed for their physicochemical characteristics according to Romanian and international standard methods, as follows: ash content (ICC 104/1), fat content (ICC 136), protein content (ICC 105/2), moisture content (ICC 110/1), falling number (ICC 107/1), gluten content (SR 90:2007) and gluten deformation index (SR 90:2007).

### 4.2. Dough Rheological Properties

#### 4.2.1. Dough Rheological Properties during Mixing and Extension

Consistograph tests (according to ICC 171 and to AACC 54-50) were performed to highlight the rheological properties of the dough during kneading. The Consistograph tests showed the following: water absorption capacity (WA), maximum pressure (PrMax), tolerance to kneading (Tol), consistency of the dough after 250 s (D250) and consistency of the dough after 450 s (D450). To determine the extension of the dough, Alveograph tests (according to ICC 121, AACC 54-30A and ISO 5530/4) were performed at a constant humidity of 14%. An Alveo Consistograph (Chopin Technologies, Cedex, France) was used for these evaluations. The Alveograph tests showed the maximum pressure (P), dough extensibility (L), swelling index (G), baking strength (W) and configuration ratio of the Alveograph curve (P/L).

#### 4.2.2. Dough Rheological Properties during Fermentation and Falling Number Values

In order to highlight the rheological properties of the dough during the fermentation stage, a Chopin Rheofermentometer (type F4, Villeneuve-La-Garenne Cedex, France) was used according to AACC89-01.01. The Rheofermentometer data obtained were for the following parameters: the total CO_2_ volume production (VT, mL), maximum height of gaseous production (H’m, mm), volume of the gas retained in the dough at the end of the test (VR, mL) and retention coefficient (CR, %). To perform these Rheofermentometer data, the dough samples were prepared as follows: 250 g of flour mix, 5 g of salt and 7 g of *Saccharomyces cerevisiae* yeast were kneaded, taking into account the water absorption value shown by the Consistograph test. Then, a falling number device (FN 1305, Perten Instruments AB Stockholm, Sweden) was used to determine the falling number values (FN, s), taking into account ICC 107/1.

#### 4.2.3. Fundamental Rheological Properties of Dough

Using the HAAKE MARS 40 Rheometer device (Termo-HAAKE, Karlsruhe, Germany), the dough fundamental rheological properties were highlighted. For these tests, the rheometer was equipped with a plate-and-plate system, 40 mm in diameter with a gap of 2 mm. The preparation of the samples for this test comprised mixing the ingredients in the Alveo Consistograph tank, taking into account the optimal value of Consistograph water absorption. Then, the samples were placed between rheometer plates and rested for 5 min for relaxation before analysis. The parameters of the frequency sweep tests were: 1 to 20 Hz, 25 °C in a previously established range of the linear viscoelasticity. The storage modulus (G′), loss modulus (G″) and loss tangent (tan δ) had been determined at a constant stress of 15 Pa for the frequency sweep tests and during heating from 25 to 100 °C at a heating rate of 4 °C per min at a frequency of 1 Hz and a fixed strain of 0.001.

### 4.3. Dough Microstructure

In order to highlight the microstructure of the dough samples, an epifluorescence light microscopy (EFLM) was used. The images were captured after the method described by us in our previously studies [49,58,72]. The EFLM used was the Motic AE 31 (Motic, Optic Industrial Group, Xiamen, PR China), equipped with LWD PH 203 catadioptric objectives (NA 0.4).

### 4.4. Bread Making

The first step in obtaining the bread samples was the dosing of the raw materials: the wheat flour and the various levels of GCF (5%, 10%, 15%, 20%) addition, the salt, the yeast and the water. The levels of the GCF addition in the wheat flour had been set by taking other studies into account, which had shown that an addition level higher than 20% GCF had a negative influence on the quality of the dough and bread [24]. Moreover, according to our data, by addition of GCF to wheat flour, the falling number decreased up to an optimum level when 20% GCF was incorporated in the dough recipe. Distilled water was used to prepare the samples according to the water absorption capacity of each flour mix. The ingredients were then kneaded for 15 min using a heavy-duty mixer (Kitchen Aid, Whirlpool Corporation, Benton Harbor, MI, USA). After the kneading stage, the dough was divided into three pieces of 400 g each. The dough samples were then fermented for 60 min at 30 °C using a fermentation chamber (PL2008, Piron, Italy). The samples were baked for 30 min, at a temperature of 220 °C, using an electrical bakery convection oven with steam production, ventilation and humidification (PF8004 D, Piron, Italy). After baking, the samples were cooled accordingly and then subjected to specific determinations. The control sample was prepared in the same way as the other samples, but without the addition of GCF in the bread recipe. The amount of water added was the optimum one for the mixed flours used according to the Consistograph data from Table 1 (water absorption value).

### 4.5. Bread Quality Evaluation

#### 4.5.1. Bread Physical Characteristics

In order to highlight the influence of the GCF addition on the quality of the bread, the specific volume was determined, applying the seed displacement method. The loaf volume, porosity and elasticity of the bread samples were determined by using the SR 90: 2007 standard method.

#### 4.5.2. Bread Color Parameters

In order to highlight the influence of the GCF addition on the color of the bread samples, the Konica Minolta CR-400 colorimeter (Tokyo, Japan) was used and the values for crumb and crust color were determined. The determination was made in CIE Lab* color system. The field of absorption of electromagnetic radiation was UV-VIS. The color parameters determined were L* (darkness/brightness), a* (shade of red/green) and b* (shade of blue/yellow).

#### 4.5.3. Texture Profile Analysis

In order to highlight the textural properties of the bread samples, the TVT-6700 texturometer device (Perten Instruments, Hägersten, Sweden) was used, which was equipped with a 10 kg-load cell. For texture determinations, 50 mm slices were cut from the bread samples. These samples were subjected to two compression procedures, up to 20% of their initial height. For this purpose, a cylindrical probe of 45 m, at a speed of 1.0 mm/s, a trigger force of 5 g and a recovery period between compressions of 15 s were used. Using the texturometer, the values for the following parameters were determined: the firmness, gumminess, cohesiveness, resilience and chewiness.

#### 4.5.4. Crumb Microstructure

The Motic SMZ-140 stereo microscope (Motic, Xiamen, China) with a 20× objective, to a resolution of 2048 × 1536 pixels was used in order to highlight the microstructure of the crumb of the control sample and of the samples with different levels of GCF addition.

#### 4.5.5. Sensory Analysis

A 9-point hedonic scale was used to highlight how the evaluators perceived the samples, from a sensory point of view. The sensory characteristics evaluated were appearance, color, aroma, taste, smell, texture and global acceptability. A panel of 20 semi-trained judges participated in the sensory determination. The judges used were previously tested for their sensory acuity according to international standards ISO8586-1, ISO8586-2 and ISO 3972 in a sensory lab from Stefan cel Mare University. The sensory analysis method used in this study was the one described by García-Gómez et al. [73]. The method was preferential, in which 1 means “extremely dislike”, 5 means “neither like nor dislike”, and 9 means “extremely like”.

#### 4.5.6. Statistical Analysis

The data were expressed as the mean ± standard deviation. Additionally, the data obtained were processed using the Statistical Package for Social Science statistical package (v.16, SPSS, Chicago, IL, USA). For all data obtained, a one-way analysis of variance (ANOVA) was used, with Tukey’s test, to determine the significant differences at a 5% level.

## 5. Conclusions

Germinated chickpea flour (GCF) can be successfully added to wheat flour in order to improve the technological process of bread making and bread quality. The determinations carried out showed that the GCF addition in different proportions influenced both the rheological properties of the dough and the quality characteristics of the final product. Regarding the rheological properties of the dough samples, it was noticed that the GCF addition led to a decrease in the values of the dough consistency, extensibility, index of swelling and baking strength. The dynamic rheological data shows that GCF addition increased the tan δ values which indicated a less elastic structure for the dough samples. The temperature-based data show that GCF addition led to a decrease in the dynamic modules. The falling number values decreased with increased GCF addition, which may indicate an increase in the α-amylase activity of the mixed flours. The Rheofermentometer data showed that the values for the maximum height of gaseous production parameter and the total CO_2_ volume production parameter improved up to the GCF addition of 10%, probably due to the increased activity of the amylase from the dough system. Regarding the quality of the bread, it was observed that the specific volume, porosity and elasticity were improved, up to the GCF addition of 15% to the wheat flour. The color parameters of the bread samples with GCF addition, compared with the control sample, had a darker color. The images captured with the stereomicroscope highlighted the fact that the crumb of the bread samples was characterized by a uniform structure of pore size, at low values of GCF addition in the bread recipe. At higher GCF addition levels, the pore size of the bread crumb increased. Generally, the bread was improved by GCF addition to the bread recipe.

## Figures and Tables

**Figure 1 plants-11-01225-f001:**
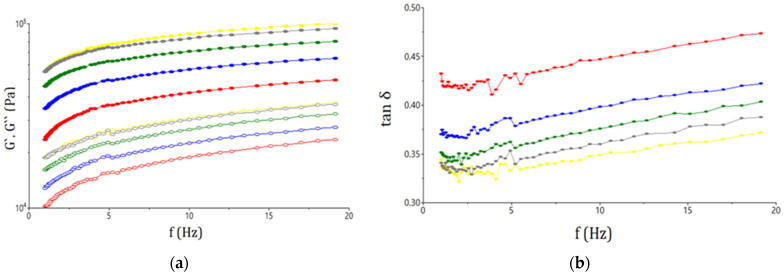
Evaluation with frequency of G′ (open symbols), G″ (solid symbols) (**a**) and tan δ (**b**) for the samples with different levels (-●-0%; -●-5%; -●-10%; -●-15%; -●-20%) of germinated chickpea flour additions.

**Figure 2 plants-11-01225-f002:**
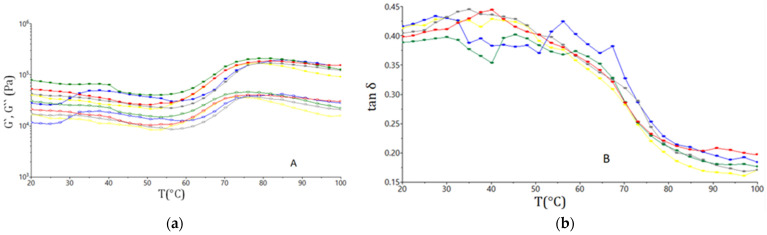
Evaluation with temperature of the G′ values (open symbols), G′′ (solid symbols) (**a**) and tan δ (**b**) for the samples with different levels (-●-0%; -●-5%; -●-10%; -●-15%; -●-20%) of germinated chickpea flour additions.

**Figure 3 plants-11-01225-f003:**
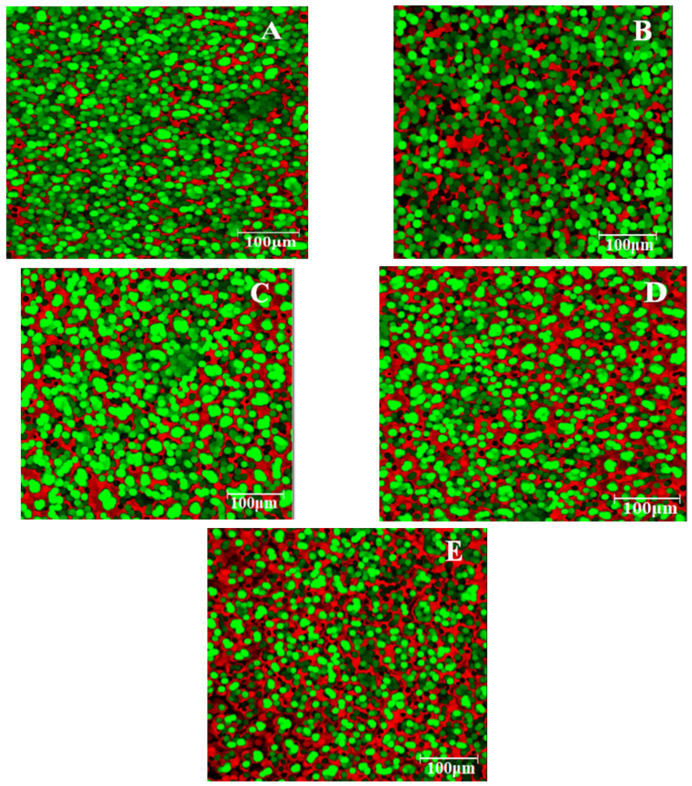
Microstructure taken by EFLM of wheat dough with germinated chickpea flour (GCF) at different levels: 0% (**A**), 5% (**B**), 10% (**C**), 15% (**D**), 20% (**E**). Red—protein; green—starch granules.

**Figure 4 plants-11-01225-f004:**
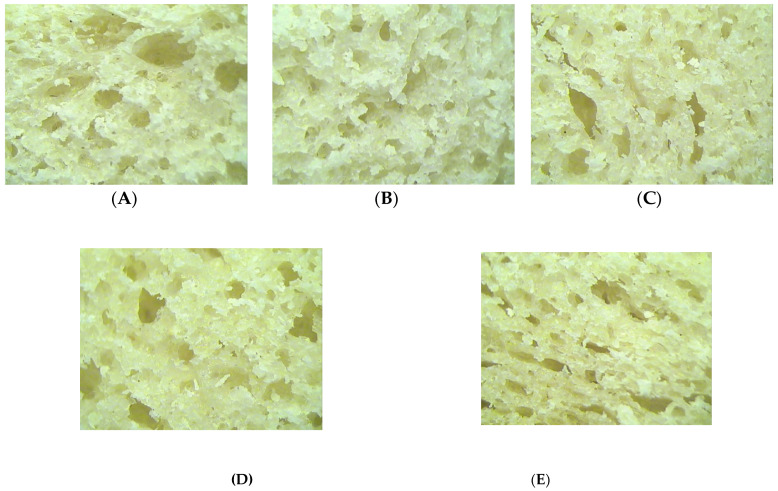
Structure of wheat dough with germinated chickpea flour (GCF) at different levels: 0% (**A**), 5% (**B**), 10% (**C**), 15% (**D**) and 20% (**E**).

**Figure 5 plants-11-01225-f005:**
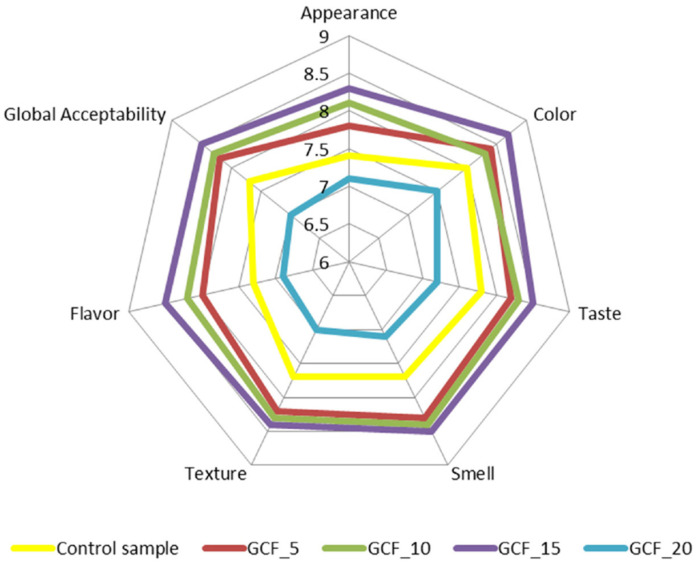
Sensory analysis for bread samples.

**Table 1 plants-11-01225-t001:** Consistograph data of dough samples with different levels of GCF addition in wheat flour.

Dough Samples	WA (%)	Tol (s)	D250 (mb)	D450 (mb)
Control	54.3 ± 0.10 ^d^	214 ± 1.00 ^b^	394 ± 2.00 ^d^	943 ± 1.00 ^e^
GCF_5	53.9 ± 0.10 ^c^	223 ± 2.00 ^c^	289 ± 5.00 ^c^	931 ± 4.00 ^d^
GCF_10	53.6 ± 0.06 ^c^	215 ± 2.00 ^b^	225 ± 3.00 ^a^	785 ± 4.00 ^b^
GCF_15	52.7 ± 0.10 ^b^	209 ± 2.00 ^b^	218 ± 1.00 ^a^	754 ± 1.00 ^a^
GCF_20	52.3 ± 0.20 ^a^	200 ± 5.50 ^a^	242 ± 3.00 ^b^	881 ± 4.00 ^c^

WA—water absorption; Tol—tolerance to mixing; D250—dough consistency after 250 s; D450—dough consistency after 450 s. The results are the mean ± standard deviation (*n* = 3). Dough samples containing germinated chickpea flour—GCF; a–e—mean values in the same column followed by different letters are significantly different (*p* < 0.05).

**Table 2 plants-11-01225-t002:** Alveograph data of dough samples with different levels of GCF addition in wheat flour.

Dough Samples	P (mm)	L (mm)	G (mm)	W (10^−4^ J)	P/L
Control	104 ± 2.51 ^a^	72 ± 1.15 ^d^	19.4±0.28 ^d^	301 ± 5.13 ^e^	1.43 ± 0.05 ^a^
GCF_5	108 ± 1.00 ^ab^	55 ± 2.00 ^c^	16.5 ± 0.30 ^c^	215 ± 2.00 ^d^	1.96 ± 0.09 ^b^
GCF_10	106 ± 1.00 ^ab^	46 ± 2.00 ^b^	15.0 ± 0.35 ^b^	190 ± 3.00 ^c^	2.30 ± 0.08 ^c^
GCF_15	110 ± 2.00 ^b^	39 ± 1.00 ^a^	13.9 ± 0.15 ^a^	172 ± 4.00 ^b^	2.82 ± 0.12 ^d^
GCF_20	107 ± 1.00 ^ab^	38 ± 2.00 ^a^	14.3 ± 0.46 ^a^	156 ± 3.00 ^a^	2.82 ± 0.18 ^d^

P—maximum pressure; L—dough extensibility; G—index of swelling; W—baking strength; P/L—configuration ratio of the Alveograph curve. The results are the mean ± standard deviation (*n* = 3). Dough samples containing germinated chickpea flour—GCF; a–e—mean values in the same column followed by different letters are significantly different (*p* < 0.05).

**Table 3 plants-11-01225-t003:** Rheofermentometer and falling number data of dough samples with different levels of GCF addition to wheat flour.

Dough Samples	H’m (mm)	VT (mL)	VR (mL)	CR (%)	FN (s)
Control	65.9 ± 0.30 ^b^	1532 ± 2.51 ^b^	1228 ± 2.51 ^c^	80.1 ± 0.50 ^d^	350 ± 3.29 ^e^
GCF_5	71.3 ± 0.20 ^c^	1622 ± 6.00 ^c^	1260 ± 3.00 ^d^	77.7 ± 0.06 ^b^	329 ± 2.00 ^d^
GCF_10	72.2 ± 0.10 ^d^	1762 ± 5.00 ^e^	1208 ± 3.51 ^b^	68.5 ± 0.06 ^a^	318 ± 4.00 ^c^
GCF_15	66.3 ± 0.20 ^b^	1641 ± 3.00 ^d^	1233 ± 2.00 ^c^	75.2 ± 0.10 ^b^	283 ± 2.00 ^b^
GCF_20	63.1 ± 0.06 ^a^	1394 ± 4.00 ^a^	1148 ± 3.00 ^a^	82.3 ± 0.45 ^e^	260 ± 1.52 ^a^

H’m—maximum height of gaseous production; VT—total CO_2_ volume production; VR—volume of the gas retained in the dough at the end of the test; CR—retention coefficient; FN—falling number value. The results are the mean ± standard deviation (*n* = 3). Dough samples containing germinated chickpea flour—GCF; a–e—mean values in the same column followed by different letters are significantly different (*p* < 0.05).

**Table 4 plants-11-01225-t004:** Physical data of bread samples with different levels of GCF addition in wheat flour.

Bread Samples	Specific Volume (cm^3^/100 g)	Porosity (%)	Elasticity (%)
Control	331.5 ± 0.74 ^b^	67.4 ± 0.86 ^b^	91.3 ± 0.57 ^b^
GCF_5	340.3 ± 0.80 ^c^	70.2 ± 1.17 ^c^	92.0 ± 0.12 ^b,c^
GCF_10	350.2 ± 1.07 ^d^	72.4 ± 0.51 ^d^	93.1 ± 0.28 ^c^
GCF_15	366.9 ± 0.71 ^e^	73.8 ± 0.73 ^d^	94.6 ± 0.53 ^d^
GCF_20	328.5 ± 0.57 ^a^	63.8 ± 0.61 ^a^	88.6 ± 0.51 ^a^

The results are the mean ± standard deviation (*n* = 3). Bread samples containing germinated chickpea flour—GCF: a–e—mean values in the same column followed by different letters are significantly different (*p* < 0.05).

**Table 5 plants-11-01225-t005:** Color data of bread samples with different levels of GCF addition in wheat flour.

Bread Samples	Crust Color			Crumb Color		
	L*	a*	b*	L*	a*	b*
Control	76.25 ± 0.94 ^d^	3.44 ± 0.27 ^a^	3.14 ± 0.43 ^a^	66.37 ± 0.88 ^d^	−4.62 ± 0.32 ^a^	1.69 ± 0.22 ^a^
GCF_5	75.38 ± 1.00 ^d^	4.87 ± 0.17 ^b^	9.07 ± 0.28 ^b^	63.17 ± 0.18 ^c^	−3.95 ± 0.05 ^b^	4.21 ± 0.23 ^b^
GCF_10	72.92 ± 0.22 ^c^	6.44 ± 0.12 ^c^	11.03 ± 0.08 ^c^	60.90 ± 0.23 ^b^	−2.52 ± 0.14 ^c^	6.35 ± 0.10 ^c^
GCF_15	68.22 ± 0.22 ^b^	9.15 ± 0.18 ^d^	12.63 ± 0.45 ^d^	58.22 ± 0.31 ^a^	−1.40 ± 0.06 ^d^	9.22 ± 0.18 ^d^
GCF_20	58.91 ± 0.82 ^a^	10.16 ± 0.24 ^e^	18.16 ± 0.20 ^e^	57.34 ± 0.38 ^a^	−0.48 ± 0.02 ^e^	11.29 ± 0.17 ^e^

The results are the mean ± standard deviation (*n* = 10). Bread samples containing germinated chickpea flour—GCF: a–e—mean values in the same column followed by different letters are significantly different (*p* < 0.05). L*: lightness; a*: green–red (−a = green and +a = red) opponent colors; b*: blue–yellow (−b = blue and +b = yellow) opponent colors.

**Table 6 plants-11-01225-t006:** Texture data of bread samples with different levels of GCF addition in wheat flour.

Bread Samples	Firmness (*N*)	Gumminess (*N*)	Cohesiveness (Adimensional)	Resilience (Adimensional)
Control	9.01 ± 3.06 ^a^	7.23 ± 1.73 ^b^	0.82 ± 0.03 ^d^	1.72 ± 0.04 ^d^
GCF_5	12.57 ± 0.39 ^ab^	6.41 ± 0.50 ^ab^	0.49 ± 0.02 ^a^	1.68 ± 0.04 ^cd^
GCF_10	16.38 ± 0.37 ^bc^	5.24 ± 0.10 ^ab^	0.67 ± 0.04 ^c^	1.58 ± 0.02 ^c^
GCF_15	19.20 ± 0.06 ^cd^	4.20 ± 0.30 ^a^	0.60 ± 0.01 ^b^	1.40 ± 0.02 ^b^
GCF_20	20.04 ± 0.54 ^d^	6.77 ± 0.23 ^b^	0.54 ± 0.01 ^ab^	1.21 ± 0.08 ^a^

The results are the mean ± standard deviation (*n* = 3). Bread samples containing germinated chickpea flour—GCF: a–d—mean values in the same column followed by different letters are significantly different (*p* < 0.05).

## Data Availability

Not applicable.

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
