# Peer review of "The Impact of Germinated Chickpea Flour Addition on Dough Rheology and Bread Quality"

_plants, 2022, doi:10.3390/plants11091225_

Round 1
Reviewer 1 Report
Reviewer comments
This manuscript presents an interesting study in the field of cereal science. The aim of this work is to evaluate the potential of addition of germinated chickpea flour on dough rheology and final quality of bread. In general, the work is good and original and the manuscript is well writing; however, the reviewer has some comments and suggestions to improve the manuscript:
Introduction
Line 25-55: the authors have sacrificed an important part of the introduction just to describe the nutritional value of chickpeas. While the objective of the work is to evaluate the addition of germinated chickpeas on the rheology and the final quality of the breads without evaluating the nutritional quality of the breads.
It will be better to detail the potential of chickpeas as an ingredient in bread-making
After that, the authors must announce the problem related to bread making which requires the use of germinated chickpea
Line 89-92: authors’ explained the role of germinated chickpea as improvers of the nutritional quality of bread; however, the nutritional evaluation was not done in this work.
Materials and methods
Bread making
Authors must justify the choice of % of GCF addition in this section
Line 637: authors must add the reference of sensory evaluation method
How authors prepare the control bread?. They must add this, to the bread making process
Authors must add a table with composition of each formulation with value of water added
Results and discussion
This section is well detailed with sufficient data
All tables and figures are clear and represent the results
Figure 5: authors must change the colour of the graphs of control and GCF 20 because they are close.
Author Response
21 April 2022
Dear Referee,
We would like to thank the referee for the close reading and for the proper suggestions. We hope that we provide all the answers to the reviewer’s comments.
Thank you very much for the recommendations to publish our paper entitled “The Impact of Germinated Chickpea Flour Addition on Dough Rheology and Bread Quality”.
The present version of the paper has been revised according to the reviewer’s suggestions.
We uploaded the corrected version of the article for which we used the red color for the addition text.
Reviewer comments
This manuscript presents an interesting study in the field of cereal science. The aim of this work is to evaluate the potential of addition of germinated chickpea flour on dough rheology and final quality of bread. In general, the work is good and original and the manuscript is well writing; however, the reviewer has some comments and suggestions to improve the manuscript:
Response: We want to thank to the referee for the close reading of our manuscript. We completed the manuscript with more informations related to the referee questions according to his/her suggestions. Also, would like to thank the reviewer for all the comments and suggestions which have helped us to improve our paper.
Reviewer: Introduction
Line 25-55: the authors have sacrificed an important part of the introduction just to describe the nutritional value of chickpeas. While the objective of the work is to evaluate the addition of germinated chickpeas on the rheology and the final quality of the breads without evaluating the nutritional quality of the breads.
It will be better to detail the potential of chickpeas as an ingredient in bread-making
After that, the authors must announce the problem related to bread making which requires the use of germinated chickpea
Response: We would like to thank to the referee for his/her suggestions. We took into account the recommendations received and in the first part of the introduction we pointed out the potential of chickpeas as an ingredient in bread-making.
Reviewer: Line 89-92: authors’ explained the role of germinated chickpea as improvers of the nutritional quality of bread; however, the nutritional evaluation was not done in this work.
Response: We would like to the referee for his/her remark. We agree with the referee point of view and therefore because the nutritional evaluation was not done in this work, we have eliminated, from the lines 89-92, the specification that we have made regarding the role of germinated chickpea as improvers of the nutritional quality of bread.
Reviewer: Materials and methods
Bread making
Authors must justify the choice of % of GCF addition in this section
Response: In the section where we described the process of preparing the bread, we explained how we have chosen the percentages of addition, taking into account other studies in the field that have been carried out previously. Also, we taken into account the falling number value (see table 3). Since to 20% GCF addition the FN value was 260 s an optimum one for bread making we considered this value a maximum one for GCF that may be added in wheat flour in our case.
Reviewer: Line 637: authors must add the reference of sensory evaluation method
Response: We have added the reference related to the sensory analysis method that we have used in this study.
Reviewer: How authors prepare the control bread?. They must add this, to the bread making process
Response: In the section where we described the process of preparing the bread, we explained how we prepared the control sample according to referee suggestions.
Reviewer: Authors must add a table with composition of each formulation with value of water added
Response: The water used for the mix flours is the optimum one mentioned in the table 1 to the Consistograph data. Therefore another table is not necessary since we already mentioned the amount of water used in bread making in another table. To make clearer this aspect we added a references to table 1 in bread making section.
Reviewer: Results and discussion
This section is well detailed with sufficient data
All tables and figures are clear and represent the results
Response: We would like to thank to the reviewer for his/her appreciation.
Reviewer: Figure 5: authors must change the colour of the graphs of control and GCF 20 because they are close.
Response: We would like to thank to the referee for his/her appreciation. We agree with his/her point of view we changed the color of the graphs.
Sincerely,
Codină et co.

Reviewer 2 Report
The research is interesting and worthy of publication. However the English needs much revision. The authors need to spend some time with a fluent English speaking colleague in order to rephrase the sentences to allow for improved sentences. Otherwise the importance of the paper will be lost.
The images illustrating the structure of the doughs are of great significance especially in relation to the dough rheology results. How much of this change can be attributed to the germinated bioactive compounds of chickpea? How much would be related to soluble fibre components in the dough diluting the protein starch matrix?
Chickpea comoposition and nutritional utilisation has been subject to a number of manuscripts. I would suggest including these references in the introduction so as to give other examples of how chickpea ingredients may be of use in human nutriation and functional foods
Perez-Perez, L.M., Huerta-Ocampo, J.Á., Ramos-Enríquez, J.R., Ruiz-Cruz, S., Wong-Corral, F.J., Rosas-Burgos, E.C., Hernández-Ortíz, M., Encarnación-Guevara, S., Robles-García, M.A., Iturralde-García, R.D., Borboa-Flores, J. and Del-Toro-Sánchez, C.L. (2021), Interaction of the human intestinal microbiota with the release of bound phenolic compounds in chickpea (Cicer arietinum L.). Int. J. Food Sci. Technol., 56: 6497-6506.
Boukid, F. (2021), Chickpea (Cicer arietinum L.) protein as a prospective plant-based ingredient: a review. Int. J. Food Sci. Technol., 56: 5435-5444.
In addition, there have been a few references recently evaluating the effects of germination on the phenolic compounds and bioactive ingredients of seeds. It may be useful to discuss these references in the introduction as a comparison to chickpea examples.
Munarko, H., Sitanggang, A.B., Kusnandar, F. and Budijanto, S. (2021), Effect of different soaking and germination methods on bioactive compounds of germinated brown rice. Int. J. Food Sci. Technol., 56: 4540-4548.
Wu, Y., Wang, H., Wang, Y., Brennan, C.S., Anne Brennan, M., Qiu, C. and Guo, X. (2021), Comparison of lignans and phenolic acids in different varieties of germinated flaxseed (Linum usitatissimum L.). Int. J. Food Sci. Technol., 56: 196-204.
The information regarding enzyme content and the effects on WA and dough rheology is important to elaborate on see if there is a real correlation between enzyme content and the factors affecting dough rheology. Do you have comparitive levels of amylase and protease in the flour and dough samples ?
Perhaps look at the refeernece below and discuss how these levels are important in bread quality.
Yang, M., Li, N., Wang, A., Tong, L., Wang, L., Yue, Y., Yao, J., Zhou, S. and Liu, L. (2021), Evaluation of rheological properties, microstructure and water mobility in buns dough enriched in aleurone flour modified by enzyme combinations. Int. J. Food Sci. Technol., 56: 5913-5922.
Otherwise a good paper.
Author Response
21 April 2022
Dear Referee,
We would like to thank the referee for the close reading and for the proper suggestions. We hope that we provide all the answers to the reviewer’s comments.
Thank you very much for the recommendations to publish our paper entitled “The Impact of Germinated Chickpea Flour Addition on Dough Rheology and Bread Quality”.
The present version of the paper has been revised according to the reviewer’s suggestions.
We uploaded the corrected version of the article for which we used the red color for the addition text.
Reviewer comments
The research is interesting and worthy of publication. However the English needs much revision. The authors need to spend some time with a fluent English speaking colleague in order to rephrase the sentences to allow for improved sentences. Otherwise the importance of the paper will be lost.
Response: The authors would like to thank the reviewer for all the appreciation and suggestions he have made. We have taken into account each recommendation. We have made the English language corrections with the help of an English teacher.
Reviewer: The images illustrating the structure of the doughs are of great significance especially in relation to the dough rheology results. How much of this change can be attributed to the germinated bioactive compounds of chickpea? How much would be related to soluble fibre components in the dough diluting the protein starch matrix?
Response: We have completed the manuscript with more explanations according to the referee suggestions. We explained the overwhelming role of soluble fiber on the gluten matrix, emphasizing that during the germination process takes place the conversion of some insoluble dietary fiber into soluble dietary fiber, which coincides with an exacerbation of competition for water between soluble fiber and gluten protein.
Reviewer: Chickpea composition and nutritional utilisation has been subject to a number of manuscripts. I would suggest including these references in the introduction so as to give other examples of how chickpea ingredients may be of use in human nutrition and functional foods
Response: We have included in the introduction the suggested references, so as to provide other examples of how chickpea ingredients may be of use in human nutrition and functional foods.
Perez-Perez, L.M., Huerta-Ocampo, J.Á., Ramos-Enríquez, J.R., Ruiz-Cruz, S., Wong-Corral, F.J., Rosas-Burgos, E.C., Hernández-Ortíz, M., Encarnación-Guevara, S., Robles-García, M.A., Iturralde-García, R.D., Borboa-Flores, J. and Del-Toro-Sánchez, C.L. (2021), Interaction of the human intestinal microbiota with the release of bound phenolic compounds in chickpea (Cicer arietinum L.). Int. J. Food Sci. Technol., 56: 6497-6506.
Boukid, F. (2021), Chickpea (Cicer arietinum L.) protein as a prospective plant-based ingredient: a review. Int. J. Food Sci. Technol., 56: 5435-5444.
Reviewer: In addition, there have been a few references recently evaluating the effects of germination on the phenolic compounds and bioactive ingredients of seeds. It may be useful to discuss these references in the introduction as a comparison to chickpea examples.
Response: We have discussed these references in the introduction according to the referee suggestions.
Munarko, H., Sitanggang, A.B., Kusnandar, F. and Budijanto, S. (2021), Effect of different soaking and germination methods on bioactive compounds of germinated brown rice. Int. J. Food Sci. Technol., 56: 4540-4548.
Wu, Y., Wang, H., Wang, Y., Brennan, C.S., Anne Brennan, M., Qiu, C. and Guo, X. (2021), Comparison of lignans and phenolic acids in different varieties of germinated flaxseed (Linum usitatissimum L.). Int. J. Food Sci. Technol., 56: 196-204.
Reviewer: The information regarding enzyme content and the effects on WA and dough rheology is important to elaborate on see if there is a real correlation between enzyme content and the factors affecting dough rheology. Do you have comparative levels of amylase and protease in the flour and dough samples ?
Perhaps look at the reference below and discuss how these levels are important in bread quality.
Response: We have explained the role of enzymes in the dough matrix and their influence on the quality of the bread we have also added the reference suggested by the reviewer. More, the amylase level of our flour may be related to the Falling Number values as we mentioned in our manuscript.
Yang, M., Li, N., Wang, A., Tong, L., Wang, L., Yue, Y., Yao, J., Zhou, S. and Liu, L. (2021), Evaluation of rheological properties, microstructure and water mobility in buns dough enriched in aleurone flour modified by enzyme combinations. Int. J. Food Sci. Technol., 56: 5913-5922.
Reviewer: Otherwise a good paper.
Response: The authors would like to thank reviewers for their appreciations and for all the suggestions because these helped us to correct our paper and to optimize it.
Sincerely,
Codină et co.
